# Deriving cloud droplet number concentration from surface based remote sensors with an emphasis on lidar measurements.

Gerald G. Mace[1]

[1.] Department of Atmospheric Sciences, University of Utah, Salt Lake City Utah, USA

*Correspondence to*: Gerald G. Mace (jay.mace@utah.edu)

**Abstract**. Given the importance of constraining cloud droplet number concentrations ($N_d$) in low-level clouds, we explore two methods for retrieving $N_d$ from surface-based remote sensing that emphasize the information content in lidar measurements. Because $N_d$ is the zeroth moment of the droplet size distribution (DSD), and all remote sensing approaches respond to DSD moments that are at least two orders greater than the zeroth moment, deriving $N_d$ from remote sensing measurements has significant uncertainty. At minimum, such algorithms require extrapolation of information from two other measurements that respond to different moments of the DSD. Lidar, for instance, is sensitive to the second moment (cross-sectional area) of the DSD, while other measures from microwave sensors respond to higher-order moments. We develop methods using a simple lidar forward model that demonstrates that the depth to the maximum in lidar attenuated backscatter ($R_{max}$) is strongly sensitive to $N_d$ when some measure of the liquid water content vertical profile is given or assumed. Knowledge of $R_{max}$ to within 5 m can constrain $N_d$ to within several 10's of percent. However, operational lidar networks provide vertical resolutions of >15 m, making a direct calculation of $N_d$ from $R_{max}$ very uncertain. Therefore, we develop a Bayesian optimal estimation algorithm that brings additional information to the inversion such as lidar-derived extinction and radar reflectivity near cloud top. This statistical approach provides reasonable characterizations of $N_d$ and effective radius ($r_e$) to within approximately a factor of 2 and 30%, respectively. By comparing surface-derived cloud properties with MODIS satellite and aircraft data collected during the Marcus and Capricorn 2 campaigns, we demonstrate the utility of the methodology.

**Short Summary**: The number of cloud droplets per unit volume, $N_d$, in a cloud is important for understanding aerosol-cloud interaction. In this study we develop techniques to derive cloud droplet number concentration from lidar measurements combined with other remote sensing measurements such as cloud radar and microwave radiometer. We show that deriving $N_d$ is very uncertain although a synergistic algorithm seems to produce useful characterizations of $N_d$ and effective particle size.

## 1 Introduction

The number of cloud droplets per unit volume ($N_d$) is essential for characterizing cloud properties. Particularly for lower tropospheric liquid-phase clouds, $N_d$ forms a bridge between atmospheric aerosol and the earth's albedo by determining how condensed water is partitioned into droplet surface area. Higher droplet concentrations for a given condensed mass result in more surface area and more reflective clouds (Twomey, 1974). Thus, many cloud parameterizations used in models include $N_d$ as one of the moments in multi-moment cloud schemes where the other moment is typically related to the mass mixing ratio (Gettelman and Morrison, 2015; Thompson and Eidhammer, 2014; Seifert and Beheng, 2005). Conceptually, using $N_d$ as a baseline parameter makes sense since droplets typically condense on hygroscopic aerosol particles (hereafter cloud condensation nuclei or CCN), thereby fixing $N_d$ as the water droplets grow in an updraft. The initial $N_d$ at the cloud base would be an upper limit on $N_d$ in the ascending updraft because coalescence processes would reduce $N_d$, and precipitation would further scavenge cloud droplets.

However, aircraft observations often show that for shallow clouds of less than 1 km in depth with minimal precipitation, $N_d$ is reasonably constant with height (Miles et al., 2000) and strongly correlated with CCN (McFarquhar et al., 2021).

In this paper, we revisit the methodology used in Mace et al., (2021; Hereafter M21) and attempt to extend that methodology with a focus on lidar measurements from below cloud. In M21, a method derived therein was applied to non-precipitating clouds where the layer-averaged radar reflectivity provided the primary source of information. While M21 used the lidar measurements at the cloud base to contribute to the first guess, M21 did not fully exploit the information content available in the lidar measurements. Here, we more thoroughly examine what the lidar signal near the cloud base can tell us about cloud properties in optically thick boundary layer clouds. Because the lidar backscatter is much larger at the cloud base than in sub-cloud drizzle, we apply the methodology to lightly precipitating and non-precipitating clouds.

## 2 Methods

### 2.1. Instruments and Comparison Approach

We focus on data collected during the summer of 2018 from two ship-based campaigns on the Australian Research Vessel (RV) Investigator and the Australian Ice Breaker Aurora Australis during voyages between Hobart, Australia, and East Antarctica. These campaigns are known respectively as the second Clouds Aerosols Precipitation Radiation and Atmospheric Composition Over the Southern Ocean (Capricorn 2) and Measurements of Aerosols, Radiation, and Clouds over the Southern Ocean (Marcus). These campaigns and a detailed accounting of instrumentation is described in McFarquhar et al. (2021) and Mace et al., (2021).

The key observations we focus on in this paper are vertically-pointing depolarization elastic-backscatter lidars, vertically pointing W-band radars, microwave radiometers, and ancillary measurements provided by radiosondes and surface meteorological instruments. This combination of active and passive instruments (radar, lidar, and radiometer) have become common in many cloud- and precipitation-focused field campaigns and enable derivation of cloud properties as we describe herein. One important synergy in this instrument suite is that the lidar is very sensitive to the droplets at cloud base while the radar is most sensitive to the cloud top region where the droplets are largest in non-precipitating clouds thereby providing immediate information on cloud layer depth. In terms of the measurable quantities, the lidar attenuated backscatter measurement is sensitive to the second moment of the droplet size distribution (DSD), the radar to the sixth moment of the DSD, and the microwave radiometer to the integrated condensed mass in the vertical column (liquid water path or LWP, hereafter). For non-precipitating clouds, Frisch et al. (1995) illustrate how the radar reflectivity profile, being proportional to the square of the condensed mass, can be cast as a weighting

function to vertically distribute the LWP. The lidar then, being the most sensitive to the smaller droplets that compose the DSD, provides information regarding how the mass is distributed into the droplets. Combining the layer depth information and the LWP, we have immediate and critical information regarding the degree of adiabaticity of the layer (Albrecht et al., 1990). We seek to exploit these synergies in the algorithms described in the following sections.

While we focus on the information in surface-based measurements, we also take advantage of airborne in situ measurements and measurements provided by satellites. Again, with a theme of synergy, in situ data provide a direct measure of the cloud properties we seek to infer from remote sensing measurements in unique and rare instances of coordination while the satellite data provide regional observations from frequently occurring overpasses. The satellite overpasses over periods of weeks to months provide

good coverage of diverse cloud fields collected over the course of a campaign. We make use of these additional platforms to both validate our algorithms but also to provide context and understanding of the processes at work in a particular cloud field.

## 2.2 Theory and Assumptions

The observed lidar attenuated backscatter $\beta_{obs}$ can be combined with other measurements to derive $N_d$ in fully attenuating liquid phase clouds when measured from the surface. Even though light precipitation may be present, we assume that $\beta_{obs}$ is dominated by a droplet distribution ($N(D)$) describable by a modified gamma function. Following Appendix B in Posselt and Mace (2014):

$$\frac{dN(D)}{dD} = N_0 \left(\frac{D}{D_0}\right)^\alpha exp\left(-\frac{D}{D_0}\right) \qquad (1)$$

Where $\frac{dN(D)}{dD}$ is the droplet number concentration per unit size D with units of cm$^{-4}$ in the cgs unit system. $N_0$ with units of cm$^{-4}$, $D_0$ with units of cm, and $\alpha$ (unitless) are respectively the characteristic number, diameter and the shape parameter of the $N(D)$ distribution function. This simple integrable function allows us to express the microphysical quantities, $N_d$, $q$ (liquid water content), $r_e$ (effective radius), $\sigma$ (extinction), and $Z$ (radar reflectivity in the Rayleigh limit), with the following expressions by integrating over all $D$,

$$N_d = N_0 D_0 \Gamma(\alpha + 1) \qquad (2)$$

$$q = \rho \frac{\pi}{6} N_0 D_0^4 \, \Gamma(\alpha + 4) \qquad (3)$$

$$r_e = \frac{D_0}{2}(\alpha + 3) \qquad (4)$$

$$\sigma = \frac{\pi}{2} N_0 D_0^3 \, \Gamma(\alpha + 3) \qquad (5)$$

$$Z = N_0 D_0^7 \, \Gamma(\alpha + 7) \qquad (6)$$

Where $\rho$ is the density of liquid water and $\Gamma$ is the gamma function. $r_e$ is derived as the ratio of the 3$^{rd}$ moment of $N(D)$ to the 2$^{nd}$ moment of $N(D)$ followed by application of the recursion relationship of the gamma function. For $\sigma$, we assume that the extinction efficiency can be approximated as 2 for integrations over typical water droplet distributions. The radar reflectivity $Z$ is written as the sixth moment of the DSD consistent with the Rayleigh approximation which is valid for cloud droplets and radar wavelengths up to W-Band (~94 GHz or ~3mm wavelength). Conversion from conventional units of mm$^6$ m$^{-3}$ to units in the cgs system (cm$^3$) requires multiplication of $Z$ by 10$^{-12}$. Using Eqns. 2-6, we develop relationships among the variables:

$$N_d = \frac{3}{4} \frac{1}{k\pi\rho} \frac{q}{r_e^3} \qquad (7)$$

$$Z = q r_e^3 C \qquad (8)$$

$$\sigma = \frac{3}{2\rho} \frac{q}{r_e} \qquad (9)$$

Where $k = \frac{(\alpha+2)(\alpha+1)}{(\alpha+3)^2}$, and $C = \frac{48\Gamma(\alpha+7)}{\pi\Gamma(\alpha+4)(\alpha+3)^3}$. Eqn. 9 was first derived by Stephens (1978) and illustrates a pathway to deriving $N_d$ from multi spectral satellite reflectance measurements. For instance, the bi spectral method applied to MODIS (Nakajima and King, 1990; Platnick et al. 2003) returns measurements of optical depth ($\tau$) and $r_e$. Since $\tau$ is the vertical integral of $\sigma$, Eqn. 3 can

be adapted for use with satellite retrievals.  A full derivation and error analysis of deriving $N_d$ and other quantities from bi spectral satellite retrievals is presented in Grosvenor et al. (2018; Hereafter G18).

Following Platt (1977) and extending through the work of Hu et al., (2007) and Li et al. (2011) among others, we express the observed lidar attenuated backscatter as

$$\beta_{obs}(R) = \beta(R)e^{-2\int \eta\sigma dR} \quad .$$ (10)

$\beta_{obs}$ is the result of 2-way attenuation through the cloud to a point R (range) in the layer and $\sigma$ is the extinction coefficient with units of inverse length where $\sigma$ is expressed in terms of the lidar ratio, $S = \frac{\sigma}{\beta}$.  A factor $\eta$ hereafter referred to as the multiple scattering factor accounts for the addition of photons to the observed signal due to multiple scattering in optically dense clouds. Defining the layer-integrated total attenuated backscatter as $\gamma = \int \beta_{\parallel+\perp}$ and the layer integrated depolarization ratio as $\delta = \frac{\int \beta_\perp}{\int \beta_{\parallel+\perp}}$ we express $\eta = \left(\frac{1-\delta}{1+\delta}\right)^2$ (Hu et al. 2009).  Platt et al. (1999) relates S with $\eta$ according to $S\eta = \frac{1-T^2}{2\gamma}$ and where T is the layer transmittance.  When the layer is fully attenuating (T=0) and $S = \frac{1}{2\eta\gamma}$.

Figure 1 illustrates two examples of $\beta_{obs}$ profiles measured by a Micropulse lidar (Lewis et al., 2020) on board the Aurora Australis during MARCUS.  Note that the units of the lidar signal in Fig. 1 are expressed as normalized relative backscatter (NRB) in Figure 1 that is equivalent to $\beta_{obs}$ via a calibration constant.  We convert NRB to $\beta_{obs}$ using a calibration technique described in O'Connor et al. (2004). We see the typically small $\beta_{obs}$ below the cloud that is due to aerosol and molecular scattering in Fig. 1a, while in Figure 1b, there is a contribution from drizzle (observed by a collocated w-band radar, not shown). There is an immediate increase in $\beta_{obs}$ at a height where condensed liquid water droplets near the cloud base activate, grow rapidly with height, and begin to dominate the lidar signal scattering.   $\beta_{obs}$ then increases exponentially according to Eqn. 4 until the two-way attenuation causes $\beta_{obs}$ to reach a maximum value, which decays exponentially.  We define the range from cloud base to the maximum in $\beta_{obs}$ as $R_{max}$.   Beyond $R_{max}$, $\beta_{obs}$ gains more contribution by multiple-scattered light depending on the lidar field of view and, in liquid clouds, the signal  becomes increasingly depolarized relative to the transmitted signal because the orientation of the electric field vector is modified by the directionality of each scattering event. The progressive depolarization of the scattered signal is a function of the droplet size distribution (Hu et al., 2009).  The overall result is quantified by η which is a factor less than 1 that effectively adds signal to $\beta_{obs}$ in Eqn. 10.  The logarithmic decay of $\beta_{obs}$ was shown by Li et al. (2011) to be related to $\sigma$:

$$\eta\sigma = -\frac{\ln\beta(R_2)_{obs} - \ln\beta(R_1)_{obs}}{2(R_2 - R_1)}$$ (11)

Where $(r_2 - r_1)$ is the range over which the change in $\beta_{obs}$ is calculated.  Because we have estimated η from measurements, we can estimate σ in the optically thick part of the layer beyond the peak in $\beta_{obs}$ using linear regression.  Li et al. (2011) compare σ derived from this method to estimates of σ derived from passive reflectances and find an uncertainty of ~13% although we assume it to be higher (20%) below.  This method's accuracy depends on calculating the rate at which the signal decays with depth in the layer.  In practice, we fit a regression line to $\beta_{obs}$ at ranges beyond $R_{max}$ until the signal is a factor of 2 above the lidar noise floor. We determine the lidar noise level from the mean $\beta_{obs}$ well beyond the point of full attenuation in the cloud layer.  The goodness of the linear regression fit depends on the number of measurements in this range where the signal is decaying. The accuracy depends on the vertical resolution of the lidar measurements for a given σ.  The accuracy of the fit is tracked and used to estimate uncertainty.

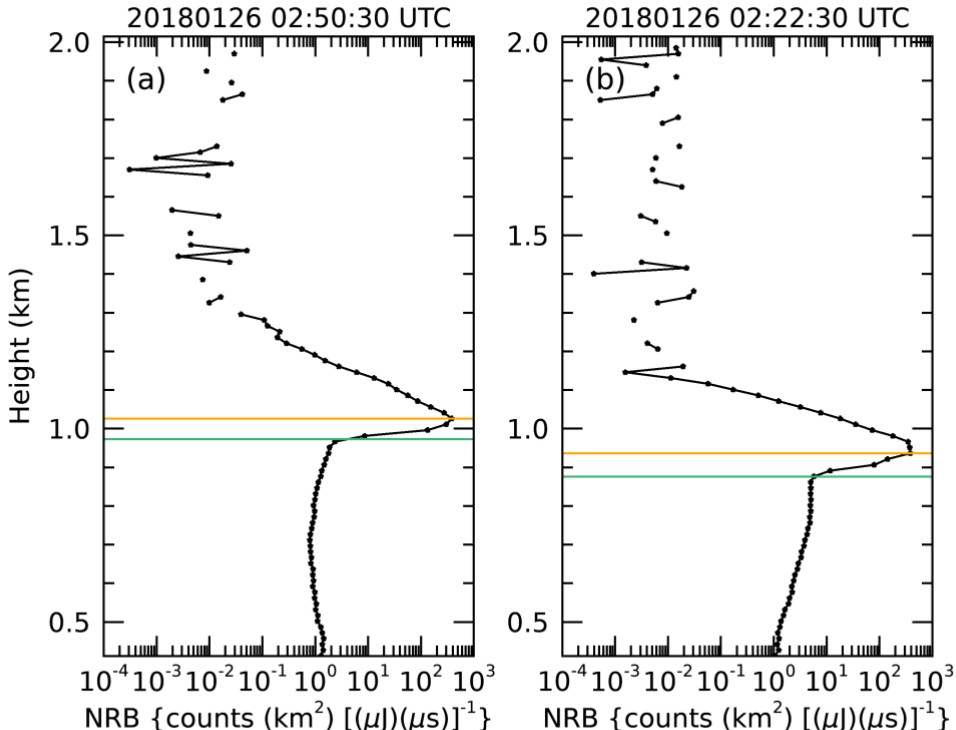

**Figure 1. Two Examples of Normalized Relative Backscatter (NRB; Campbell et al., 2002) from Micropulse Lidar data collected during Marcus on January 26, 2018. a) shows a profile in a non-drizzling cloud collected at 02:50:30 UTC. b) shows a profile collected at 02:22:30 that had sub cloud drizzle as indicated by the cloud radar. The green line indicates the height determined to be cloud base while the red line indicate the maximum in $\beta_{obs}$. The distance between the green and red lines is defined as $R_{max}$**

### 2.3 Direct Calculation of $N_d$ and $r_e$

The growth of the lidar signal from cloud base to $R_{max}$ can be used to gain information about the cloud layer. Taking the natural logarithm of both sides of Eqn. 10, recognizing that $\beta S_c = \sigma$, and then differentiating with range $r$ in the cloud layer, we can write,

$$\frac{\partial \ln \beta_{obs}}{\partial R} = \frac{\partial \ln \sigma}{\partial R} - 2\eta\sigma. \tag{12}$$

We next derive an expression relating $\sigma$ in terms of $N_d$ and $q$. We can simply express $N_0$ in terms of $N_d$: $N_0 = \frac{N_d}{D_0\Gamma(\alpha+1)}$, and substitute into Eqn 5:

$$\sigma = \frac{\pi}{2}\frac{N_d}{\Gamma(\alpha+1)}D_0^2\,\Gamma(\alpha+3) \tag{13}$$

Then solve the expression for $\sigma$ in Eqn. 5 for $D_0$: $D_0^3 = \frac{\sigma}{\frac{\pi}{2}N_0\Gamma(\alpha+3)}$ and substitute into Eqn. 3 and rearrange to obtain $D_0 = \frac{3}{\rho}\frac{q}{\sigma}\frac{\Gamma(\alpha+3)}{\Gamma(\alpha+4)}$. Now substitute the expression for $D_0$ into Eqn. 13 and rearrange:

$$\sigma = N_d^{\frac{1}{3}}q^{\frac{2}{3}}\mathrm{B}. \tag{14}$$

where $B = \left(\frac{9\pi}{2\rho^2}\frac{[\Gamma(\alpha+3)]^3}{[\Gamma(\alpha+4)]^2\Gamma(\alpha+1)}\right)^{\frac{1}{3}}$ collects constants and assumptions. Now we combine Eqns. 12 and 14: $\frac{dln(\beta_{obs})}{dr} =$

$\frac{BN_d^{\frac{1}{3}}}{\sigma}\frac{d}{dr}\left(q^{\frac{2}{3}}\right) - 2\eta\sigma$. Solving the derivative for $q$,

$\frac{d}{dr}\left(q^{\frac{2}{3}}\right) = \frac{2}{3}q^{-\frac{1}{3}}\frac{dq}{dr}$, substituting into Eqn. 14 and simplifying we arrive at: $\frac{dln(\beta_{obs})}{dr} = \frac{2}{3}\frac{dlnq}{dr} - 2B\eta N_d^{\frac{1}{3}}q^{\frac{2}{3}}$ and solving for $N_d$:

$$N_d^{\frac{1}{3}} = \frac{\frac{2}{3}\frac{dlnq}{dR} - \frac{dln(\beta_{obs})}{dr}}{2B\eta q^{\frac{2}{3}}} \tag{15}$$

Since $q = f_{ad}\Gamma_l R$ where $\Gamma_l$ is the layer mean adiabatic liquid water lapse rate that depends on temperature and pressure and the moist adiabatic lapse rate (G18), we substitute into Eqn. 15 and noting that $\frac{dlnq}{dR} = \frac{1}{R}$, we can write Eqn. 15 as,

$$N_d = \left(\frac{\frac{2}{3r} - \frac{d\ln\beta_{obs}}{dr}}{2\eta(\Gamma_l R f_{ad})^{\frac{2}{3}}B}\right)^3 \tag{16}$$

Now where $\frac{dln(\beta_{obs})}{dr} = 0$ at $R_{max}$, we simplify the expression to arrive at

$$N_d = \frac{1}{27B^3\eta^3\Gamma_l^2 R_{max}^5 f_{ad}^2} \tag{17}$$

In Eqn. 17, $N_d$ is a function of observable quantities with an assumption that the liquid water profile has an adiabatic shape. The DSD shape parameter $\alpha$ is also assumed and typically given a value that conforms to in situ data (see below). $f_{ad}$, which scales the adiabatic liquid water content, can be calculated as the ratio of the vertically integrated liquid water mass or liquid water path

(LWP) that is readily retrieved from measurements collected by a microwave radiometer (Turner et al., 2016) to the adiabatic LWP that can be derived by integrating $\Gamma_l$ over the depth of the layer (G18). The depth of the layer must be determined from some means such as a vertically pointing cloud radar or perhaps from recent radiosonde soundings. Thus, $N_d$ can be derived with a combination of a depolarization lidar, some means of determining cloud top, and a microwave radiometer. Neither the lidar nor the radar, if present, must be calibrated to derive $N_d$ with Eqn. 17. With LWP and $N_d$, and a measure of layer depth, it is straightforward to estimate a characteristic cloud droplet size. Typically, the cloud top $r_e$ is most representative of the layer reflectance and is derived from bi spectral measurements such as MODIS to which we will compare later. Following G18,

$$r_e = \left(\frac{\frac{3h}{4\pi\rho_l}\Gamma_l f_{ad}}{kN_d}\right)^{1/3} \tag{18}$$

where h is the layer thickness and k is the cubed ratio of a volume weighted characteristic droplet size to the effective droplet size assumed constant at 0.8 following G18.

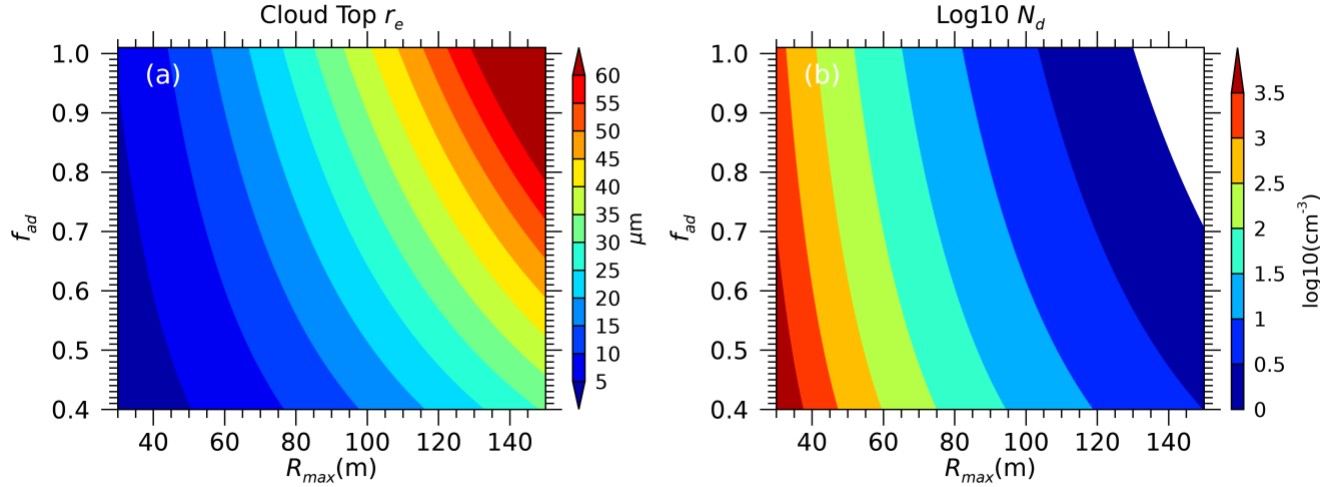

**Figure 2. Response of Equation 18 (a - $r_e$) and Equation 17 (b - $N_d$) to typical values of $R_{max}$ and $f_{ad}$.**

Figure 2 shows the response of Equations 17 and 18 to typical ranges of $R_{max}$ and $f_{ad}$. In these calculations, we fix $\eta$ at 0.4 (a typical value for the lidar on CAPRICORN 2) and the cloud layer thickness at 500 m. We find that $R_{max}$ contributes most significantly to the $N_d$ calculation, given the fifth power exponent in the denominator of Eqn. 17. $N_d$ ranges from near 1000 cm$^{-3}$ for low $R_{max}$ values that would correspond to very opaque layers to values less than 10 cm$^{-3}$ for layers with $R_{max}$ exceeding 100 m. These correspond to the approximate typical extremes for $R_{max}$ found in measurements. $r_e$ ranges from 5 $\mu$m for small $R_{max}$ to more than 50 $\mu$m for very large $R_{max}$ corresponding to the change in $N_d$ from high to low, respectively. For a given $R_{max}$, an increasingly adiabatic cloud layer causes $N_d$ to decrease and $r_e$ to increase. This tendency makes physical sense since for our simple conceptual model of an adiabatically increasing $q$ profile, increasing $f_{ad}$ for a given LWP and layer thickness (h) implies more liquid water in the profile. Therefore, for a given $R_{max}$, fewer and but larger droplets are required to achieve a given extinction profile that allows the lidar beam to penetrate the layer.

While Eqns. 17 and 18 produce physically plausible results as illustrated in Fig. 2, the sensitivity of $N_d$ to uncertainty in $R_{max}$ is substantial. The resulting uncertainty in $N_d$ then translates into uncertainty in $r_e$. Clearly, with the typical range in $R_{max}$ between a few 10's of meters to values not much greater than about 100 m, the vertical resolution of the lidar has a significant bearing on how well we can know $R_{max}$. Lidars in operational networks typically operate with range bin spacing of between 10 and 15 m. The Micropulse lidars operated by the DOE Atmospheric Radiation Measurement (ARM) program (Mather, 2021) use 15 m spacing while Vaisala laser ceilometers use a range bin spacing of 10 m. We use a bootstrap approach to evaluate the effect of this uncertainty in $R_{max}$. We assume that the 1 standard deviation uncertainty in $R_{max}$ would be ½ of the range bin spacing. Fixing the uncertainty in $f_{ad}$ and $\eta$ at 20% and allowing a variable $R_{max}$ uncertainty of 1m, 5m, 10m, and 15m, we use a normally distributed set of random numbers to perturb the $R_{max}, f_{ad}$, and $\eta$ about their assumed values prior to implementation of Eqns. 9 and 10. 25000 iterations are used to compute the frequency distribution of the resulting $N_d$ and $r_e$ (Fig. 3) for each $R_{max}$ uncertainty. We find that range bin spacing in excess of 10 m is inadequate for calculating $N_d$. A 30 m range bin spacing results in a normalized standard deviation in the $N_d$ distribution for the example shown here of ~3. The $r_e$ normalized standard deviation is approximately 29% in this case. The uncertainty in $N_d$ and $r_e$ decrease as the uncertainty in $R_{max}$ is reduced from 15 m to 1 m. At 1 m and 5 m uncertainty in $R_{max}$ corresponding to 2 m and 10 m range bin spacing, $N_d$ ($r_e$) has fractional uncertainties of 0.16 (0.16) and 0.55 (0.18), respectively. These levels of uncertainty would convey useful information about a cloud layer although the magnitude of the

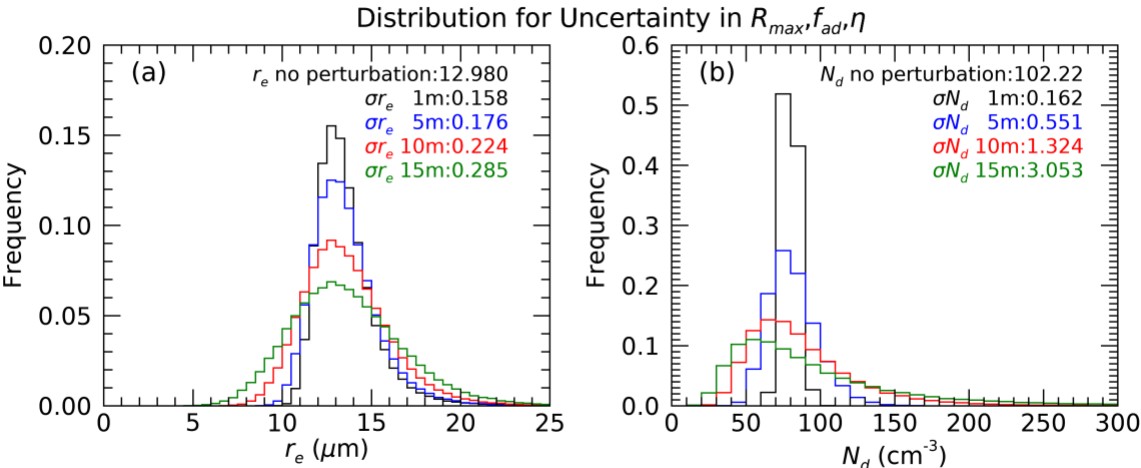

**Figure 3. Sensitivity of Eqns. 17 (a) and 18 (b) to uncertainty in input parameters. Inset lists the resulting uncertainties corresponding to the color-coded frequency distributions. Insets list normalized standard deviations for an assumed standard deviation in $R_{max}$ of 1, 5, 10, and 15 m. The lidar range gate spacing would be twice the standard deviation in $R_{max}$.**

uncertainty as illustrated by the frequency distribution (blue) in Fig. 3 illustrates the uncertainty is non negligible relative to the typical ranges of these quantities. The ranges of uncertainty that we encounter with typical operational lidars and ceilometers are only marginally to insignificantly informative.

### 2.4 An Optimal Estimation Algorithm

To lessen the effects of uncertainty in $R_{max}$, we attempt to bring additional information to bear by developing a Bayesian optimal estimation (OE) inversion algorithm (Maahn et al., 2019) to retrieve $N_d$ and $r_e$. This methodology allows us to use additional data

sources that contribute to our understanding of droplet $N_d$ and $r_e$ while balancing the observational and forward modeling errors that contribute to retrieval uncertainty. In addition to the independent variables in equations 9 and 10, we also use the layer $\sigma$ derived from the lidar data (Eqn. 5) and the radar reflectivity near cloud top ($Z_{top}$) from a collocated millimeter radar. We choose to use the radar reflectivity near the cloud top to avoid, to the extent possible, multimodal droplet distributions that often occur as drizzle or snow sediments through a cloud layer. Near layer top, at least for reasonably shallow clouds, we assume the precipitation droplet mode to be nascent and the cloud droplet distribution to be approximately unimodal. Inspection of aircraft in situ drop size distributions collected over multiple campaigns reasonably support this assumption (Lawson et al., 2017). $Z_{top}$ provides a useful constraint on the liquid water profile's shape and conveys information on $f_{ad}$ and $r_e$. We define an observational vector,

$$y = [R_{max} \quad \sigma \quad LWP \quad Z_{top}] \tag{19}$$

An observational error covariance matrix, Sy, is a 4x4 element matrix that records the uncertainty of the measurements in y due to random noise and uncertainties in forward modeling of that quantity along the diagonal. We allow for covariance among the observations using the correlations listed in Table 1 and the variances of the individual quantities as listed along the diagonal. These correlations are derived from the Capricorn 2 and Marcus combined data set. We find significant correlations among the measurements in y. These correlations show that the measurements in y are not independent and are not, therefore, unique in terms of information. We address the information content below.

**Table 1. Sources of uncertainty estimates (diagonal) and correlations (off diagonal) among measurements in y (Eqn. 11) used in the OE algorithm. Correlations are derived from the combined Marcus and Capricorn 2 data sets.**

| | $R_{max}$ | $\sigma$ | LWP | $Z_{top}$ |
|---|---|---|---|---|
| $R_{max}$ | Lidar Range Bin Space | | | |
| $\sigma$ | -0.58 | 20% (Lin et al. 2011) | | |
| LWP | +0.24 | -0.22 | 20 g m-2 (LWP<100) 30% (LWP>100) (Turner et al., 2016) | |
| $Z_{top}$ | +0.23 | +0.48 | +0.47 | 1 dB Capricorn, 4 dB Marcus (Kollias et al., 2019) |

The quantities to be estimated and their covariance are denoted in the state vector x respectively:

$$x = [N_d \quad r_e] \tag{20}$$

And $S_x$ is a 2x2 element matrix that records the uncertainties of x along the diagonal. $r_e$ is assumed to be near the layer top as defined in Eqn. 18.

We use x and additional observations and assumptions to derive a forward calculation of y or F(x) based on initial and incremental x guesses (see below) with a simple forward model. Our forward model begins with the observed thermodynamics such as temperature, pressure and relative humidity profiles, cloud base height, and layer thickness. With an observed or simulated LWP

and a temperature-dependent $\Gamma_l$, we create a vertical profile of liquid water that varies with an adiabatic shape scaled by $f_{ad}$. Using an assumed shape parameter ($\alpha=2$, justified below), we then calculate profiles of $r_e$ and $N_d$ allowing us to estimate the terms in $y$ using the simple lidar equation (Eqn. 11) and the expressions for Z and $\sigma$ in Eqns. 8 and 9.

To derive $x$ from $y$ using OE, we express the first order derivatives of y with respect to x in a Jacobian matrix, $K_x$, that has dimensions of the number of elements in y (4) by the number of elements in x (2):

$$K_x = \begin{matrix} \dfrac{\partial R_{max}}{\partial N_d} = -0.29 & \dfrac{\partial \sigma}{\partial N_d} = 0.24 & \dfrac{\partial LWP}{\partial N_d} = 0 & \dfrac{\partial Z_{top}}{\partial N_d} = 0.01 \\[2ex] \dfrac{\partial R_{max}}{\partial r_e} = 0.92 & \dfrac{\partial \sigma}{\partial r_e} = -2.9 & \dfrac{\partial LWP}{\partial r_e} = 0.44 & \dfrac{\partial Z_{top}}{\partial r_e} = 1.2 \end{matrix}$$

These terms are calculated analytically using the expressions in Eqns. 2-10, 14 and 17. Also, we set $\frac{\partial LWP}{\partial N_d} = 0$ because we assume that the amount of water made available for condensation is the result of thermodynamics while how that water is distributed into droplets depends more on the CCN that is available for the water to condense onto. The quantities listed in the $K_x$ matrix show typical values of the terms for Case 5 listed in Table 3 below in terms of $\frac{\partial \ln(y)}{\partial \ln(x)}$. We find that $r_e$ influences $\sigma$, LWP, and $Z_{top}$ in predictable ways. For instance, the derivative is negative in the $r_e$ - $\sigma$ relationship. The sensitivities of the observations in y are much more sensitive to $r_e$ than to $N_d$ illustrating the challenge of retrieving $N_d$ with remote sensing observations as discussed earlier.

The OE formalism derives x by balancing the uncertainties and information in the measurements with what is known about the statistical properties of x given the atmospheric state. The information from prior knowledge is contained in an a priori vector of

statistical estimates of the quantities in x (Eqn. 12) or $x_a$ and their covariance, $S_a$. For the prior estimate of $N_d$, we reason that coincident cloud condensation nuclei (CCN) measurements provide an upper limit on the droplet number. These measurements were collected during Marcus and CAPRICORN 2 and are available hourly when the wind direction was favorable by not contaminating aerosol inlets with ship exhaust (Humphries et al. 2021). These hourly CCN measurements collected by a Droplet Measurement Technologies (DMT) CCN-100 at 0.2% supersaturation are simply multiplied by 0.8 to account for coalescence processes and used in $x_a$. We found that the use of CCN, while a broad constraint and upper bound on $N_d$, was quite necessary for accurate convergence of the OE algorithm. The hourly standard deviation of the CCN is then used along the diagonal of $S_a$. When CCN are not available, within the previous 6 hours, we use averages of the surface-based CCN measurements for the latitudinal bands from 40°S-50°S, 50°S-60°S, and >60°S (Humphries et al., 2023). For the prior value of $r_e$, we use the 0.8*CCN, the LWP, and layer thickness in Eqn. 10. For $r_e$ we use in situ aircraft data collected during the Southern Ocean Cloud Radiation and Aerosol

Transport Experiment (SOCRATES; McFarquhuar et al., 2021) that was conducted in the Southern Ocean region south of Hobart Australia during the Austral Summer of 2018 by the NSF/NCAR HIAPER Gulfstream V (GV) aircraft. In this campaign, the GV completed 15 research flights. We combine the Cloud Droplet Probe (CDP manufactured by DMT) and 2DS (manufacture by SPEC Inc.) measurements into a single droplet size distribution (DSD) and use a moments minimization method (Zhao et al., 2011) to estimate Eqn. 1 for each low-level cloud 1-second DSD. See Baumgardner et al., (2017) and Lawson et al. (2006) for discussions of the in situ droplet probes. W-Band radar reflectivity is then calculated using Eqn. 3. For a particular retrieval where we have a measured $Z_{top}$, the Socrates data set is searched for all instances where Z is within 1.5 dB of the measurement and the prior $r_e$ is then estimated from the mean of the in-situ measurements. For the covariance among the quantities in $S_a$, we know from analysis of in situ data that $r_e$ and $N_d$ are strongly correlated (G18) at a level of ~0.7 among those terms.

The OE formalism also allows us to quantify the added uncertainty in our forward model calculations due to model parameters and assumptions (Maahn et al. 2019; Austin and Stephens, 2001) which we take to include $\alpha$ (droplet distribution function shape parameter), $f_{ad}$ (the adiabaticity of the column) and $\eta$. We find that a value of $\alpha=2$ with a standard deviation of 1.5 reasonably characterizes the in-situ cloud data collected during Socrates. $f_{ad}$ is estimated by taking the LWP and cloud thickness observations collected over the Marcus and CAPRICORN 2 voyages and deriving a linear regression of $f_{ad}$ in terms of LWP following Miller et al., (1998) to wit: $f_{ad} = 1.-(0.002 * LWP)$. With LWP in g m$^{-2}$, this equation returns $f_{ad}$=0.6 for LWP=200 gm$^{-2}$ and 0.5 for LWP=250 g m$^{-2}$. The scatter in the LWP-$f_{ad}$ observations suggest an uncertainty in this estimate of 0.15. $\eta$ is derived from the depolarization lidar data following the method described in Hu et al. (2007). While the uncertainty of this quantity is difficult to assess, examining the consistency of the estimates over periods of persistent cloud cover we determined that an uncertainty of 30% is reasonable. A term of the form $K_b S_b K_b^T$ is added to the instrumental uncertainties where $K_b$ is a Jacobian matrix that contains

the first derivatives of the measurements in y with respect to $\alpha, f_{ad}$, and $\eta$ determined through finite differences in the forward model:

$$K_b = \begin{matrix} \dfrac{\partial R_{max}}{\partial \alpha} = -0.08 & \dfrac{\partial R_{max}}{\partial f_{ad}} = -0.60 & \dfrac{\partial R_{max}}{\partial \eta} = -0.63 \\[2mm] \dfrac{\partial \sigma}{\partial \alpha} = 0.11 & \dfrac{\partial \sigma}{\partial f_{ad}} = 0.55 & \dfrac{\partial \sigma}{\partial \eta} = 0.03 \\[2mm] \dfrac{\partial LWP}{\partial \alpha} = 0.20 & \dfrac{\partial LWP}{\partial f_{ad}} = 1.0 & \dfrac{\partial LWP}{\partial \eta} = 0.0 \\[2mm] \dfrac{\partial Z_{top}}{\partial \alpha} = -0.35 & \dfrac{\partial Z_{top}}{\partial f_{ad}} = 2.0 & \dfrac{\partial Z_{top}}{\partial \eta} = 0.0 \end{matrix}$$

The numbers in the $K_b$ matrix expression are in terms of $\frac{\partial \ln (y)}{\partial \ln (x)}$ and are derived from the forward model over the physically reasonable ranges of the parameters. We find that these numbers vary by less than 20% in the Capricorn and Marcus data sets. $S_b$ contains the variance of $\alpha, f_{ad}$, and $\eta$ determined from in situ and remote sensing mearuements. We assume that the covariance among these quantities can be neglected.

Inversion of y for x then follows a standard iterative approach by applying a Gauss-Newton minimization technique derived in Rodgers (2000). See also Maahn et al., (2019). In this approach, successive guesses of x are derived using the well-known

expression,

$$\delta x = \left( S_a + K_x S_y K_x^T \right)^{-1} \left[ S_a^{-1}(\hat{x} - x_a) + K_x^T S_y^{-1}\left(y - F(\hat{x})\right) \right] \tag{21}$$

Where $\hat{x}$ is a present guess, $F(\hat{x})$ is the forward estimate of the measurements in y using the present guess. $\delta x$ then becomes the next increment on $\hat{x}$. Eqn. 21 is iterated until either a convergence criterium is met or divergence of the result occurs. Typically, less than 10 iterations are necessary if the algorithm converges which it does > 90% of the time in non-precipitating conditions while convergence occurs less frequently as drizzle and light snow increase due to the inability to accurately estimate $R_{max}$.

**2.5 Optimal Estimation Algorithm Evaluation**

The response of the OE algorithm is equivalent to the results presented in Fig. 3, except that additional information is used to

hopefully lessen the effects of uncertainty in $R_{max}$. In Table 2, we list 6 cases that we use to examine the response of the OE algorithm in terms of the retrieved quantities and their uncertainties. Cases 2, 4 and 6 differ from cases 1, 3, and 5, respectively only by the level of uncertainty applied. Cases 2, 4, and 6 use twice the listed uncertainties in Cases 1, 3, and 5 but otherwise identical inputs. Cases 1 and 2 are designed to illustrate a situation that might be found in a heavy aerosol environment with a low

$R_{max}$, high $\sigma$, and low $Z_{top}$ that produces high $N_d$, small cloud drops and moderately high LWP. Cases 3 and 4 show the opposite with a rather large $R_{max}$ and lower $\sigma$. $Z_{top}$ is set higher with a larger LWP. The algorithm returns a small $N_d$ and large $r_e$ in cases 3 and 4. Cases 5 and 6 are in between the two extremes. $f_{ad}$ in these cases range from 0.8 to 0.9, and this is by design as the cloud depth is specified. The uncertainties listed in Table 2 are used in Cases 1, 3, and 5; except for $Z_{top}$ which is listed in dB, the uncertainties are a fraction of the measurement. As a fraction of the returned values, the 1 standard deviation uncertainties do not change significantly from case to case, and they respond predictably to a doubling of the observational errors increasing approximately by a factor of 2. We also test the OE uncertainty by randomly perturbing the observations about their stated uncertainties until the error statistics converge. These are reported in Table 2 in the "Bootstrapping" column. The bootstrap experiment generally returns uncertainty in $r_e$ that is equivalent to or slightly smaller than the OE results. For $N_d$, the bootstrap experiment returns marginally larger uncertainties than the OE results.

The Shannon information content measures the extent to which the observations reduce the uncertainty in the prior. The studies of L'Ecuyer et al. (2006) and Cooper et al. (2006) provide detailed discussions of this concept. Doubling the observational uncertainty reduces the information content by approximately 1/3. The number of independent parameters is less than the number of elements in y (the observations) because the observations are correlated. For instance, as shown in Table 2, $R_{max}$ and $\sigma$ both constrain $N_d$ while LWP and $Z_{top}$ constrain $r_e$. Even in the lower error cases, the observations do not provide sufficient information to retrieve three independent quantities, suggesting that the results are correlated and not independent.

The uncertainty in $r_e$ remains roughly equivalent to the results shown in Fig. 3, although we consider the results of the OE to be more accurate because a better accounting of information is used. Notable is the magnitude of the uncertainties for the retrieved $N_d$. We find that it remains large, although the additional information provided by the other observations reduces the uncertainty compared to the results in Fig. 3. We also tested how well the OE algorithm without $R_{max}$ would do where just extinction is the primary constraint on $N_d$. This was accomplished by setting the Kx term $\frac{\partial R_{max}}{\partial N_d} = 0$. We found that for the uncertainties in the other quantities listed in Table 2, the uncertainty in $N_d$ was approximately 150%, showing that $R_{max}$ is a useful quantity in this regard. However, retrieval of $N_d$ remains highly uncertain when lidar range bin spacing exceeds 5 m.

**Table 2. Cases used to illustrate the response of the OE algorithm. Observables are listed in rows 2-5 (shaded) with uncertainties in parentheses. The retrieved quantities, their uncertainties in the OE algorithm and using a bootstrap approach (see text) are in lower rows. We also list the Shannon information content in bits, and the number of independent observations in the retrieval as derived from the OE formalism – see Rodgers, (2000). Cases 2, 4, and 6 have observational uncertainties a factor of 2 greater than Cases 1, 3 and 5.**

|  | Case 1 | Case 2 | Case 3 | Case 4 | Case 5 | Case 6 |
|---|---|---|---|---|---|---|
| $R_{max}$ (m) | 38 (4) | 38 (8) | 62 (6) | 62 (12) | 56 (5.5) | 56 (11) |
| $\sigma$ (km$^{-1}$) | 28 (4.5) | 28 (9) | 16 (2.5) | 16 (5) | 23 (3.5) | 23 (7) |
| $Z_{top}$ (dBZ) | -19 (2) | -19 (4) | -12 (2) | -12 (4) | -15 (2) | -15 (4) |
| LWP (g m$^{-2}$) | 126 (30) | 126 (60) | 101 (25) | 101 (50) | 150 (37) | 150 (74) |
| $N_d$ (cm$^{-3}$) | 229 | 231 | 36 | 37 | 95 | 91 |
| $N_d$ OE Uncert. | 0.69 | 0.83 | 0.70 | 0.84 | 0.70 | 0.84 |

| | | | | | | |
|---|---|---|---|---|---|---|
| (Fraction) | | | | | | |
| $N_d$ Bootstrap Uncert. (Fraction) | 0.77 | 0.93 | 0.88 | 1.2 | 0.95 | 1.2 |
| $r_e$ (um) | 9.8 | 9.9 | 16 | 15 | 13 | 12 |
| $r_e$ OE Uncert. (Fraction) | 0.24 | 0.42 | 0.19 | 0.40 | 0.18 | 0.40 |
| $r_e$ Bootstrap Uncert. (Fraction) | 0.23 | 0.35 | 0.28 | 0.32 | 0.27 | 0.34 |
| Info (bits) | 3.1 | 1.2 | 3.6 | 1.2 | 3.5 | 1.7 |
| # Ind Params | 1.7 | 1.4 | 1.7 | 1.4 | 1.7 | 1.4 |

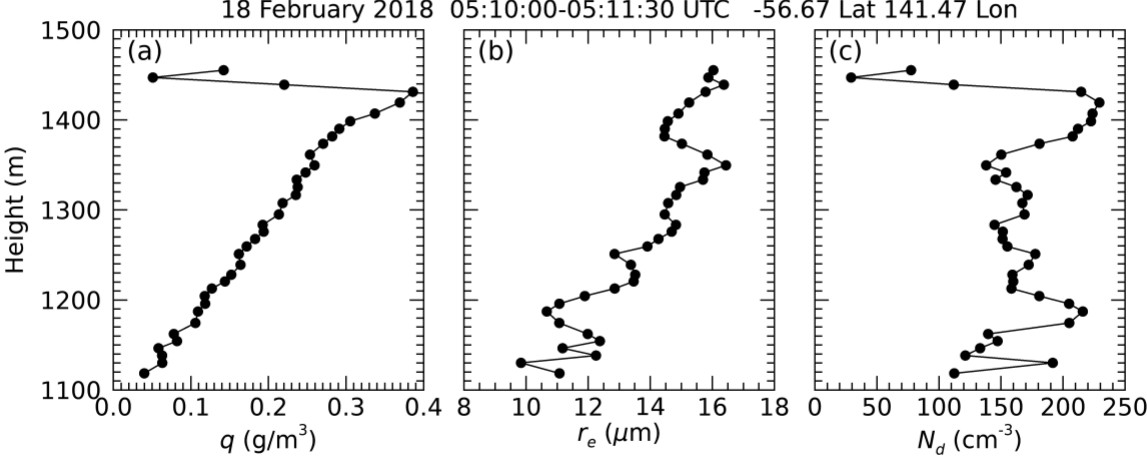

**Figure 4. Ramp through an MBL cloud layer on February 18, 2018 collected by instruments on the NCAR Gulfstream V during Socrates. This ramp was conducted near the RV Investigator ship during Capricorn 2. As a function of height between cloud base near 1100 m and cloud top near 1450m, the panel show (a) q, (b) $r_e$, and (c) $N_d$.**

To provide a more realistic evaluation of the OE algorithm performance, we use data collected during the Socrates campaign, where ramps (constant rate ascents and descents between cloud base and cloud top) through low-level cloud layers were conducted. Such a ramp is depicted in Fig. 4 which was collected on February 18, 2018 (hereafter 2/18) at 0510 UTC when the GV was conducting a mission near the R/V Investigator at 57°S and 142°E. We will expand on the February 18 case study below. For this analysis, we focused on 1-second data collected by the CDP that recorded droplet spectra in 2 $\mu$m size bins up to 50 $\mu$m. The aircraft entered the cloud layer with a temperature near -5°C at 1100 m. $q$ and $r_e$ steadily increased as the GV ascended and exited the cloud layer approximately 90 seconds later at an altitude of 1450 m where $q$ reached a maximum of 0.4 g m$^{-3}$ and the re near

cloud top was ~15 microns. We note an interesting structure in the vertical $r_e$ profile with a sudden decrease near 1375 m. During this ascent, $N_d$ was quite variable but averaged 150 cm$^{-3}$ through most of the ramp until 1375 m where there is an abrupt increase in $N_d$ to ~225 cm-3 in conjunction with the decrease in $r_e$. Integrating q vertically through the layer, the LWP was 65 g m$^{-2}$ with an adiabatic LWP of 80 g m$^{-2}$, suggesting a sub-adiabatic layer with $f_{ad}$ of ~0.8. The radar reflectivity time series (discussed later) shows that drizzle was occurring sporadically during this case. We used the cloud droplet concentrations collected during the ramp to get $R_{max}$ (32 m), the expression for Z (Eqn. 8) to estimate $Z_{top}$ (-15 dBZe), and the cross-sectional area of the droplet distribution to estimate $\sigma$ (layer mean of 30 km$^{-1}$ and layer optical depth ($\tau$) of 14). These values were used as input to the lidar forward model. We implement the OE algorithm with $f_{ad}$ and LWP to get a retrieved $N_d$ of 165 cm$^{-3}$ and $r_e$ of 14 $\mu$m in reasonable agreement with the input data.

We repeated this exercise for other ramps collected during Socrates, excluding ramps that were super-adiabatic or had non-adiabatic structure in the vertical profile, reasoning that the finite distance over which the ramps occurred (~10-20 km) had the potential to sample cloud elements of varying properties. For instance, on 2/18 three additional ramps were rejected. The observational uncertainties used in the inversion are as discussed above for Cases 1, 3, and 5. Figure 5 shows the relationship between observed and retrieved $N_d$ and $r_e$., showing that the OE algorithm can reasonably capture the characteristics of the cloud layers. While we would expect the algorithm to provide a reasonable comparison of the retrieved and observed $N_d$ and $r_e$ in this rather contrived experiment, we note that the OE uncertainty, for the most part, extends over the 1:1 line, suggesting that the characterization of uncertainty in the retrieved quantities is a reasonable estimation of the actual uncertainty of the algorithm.

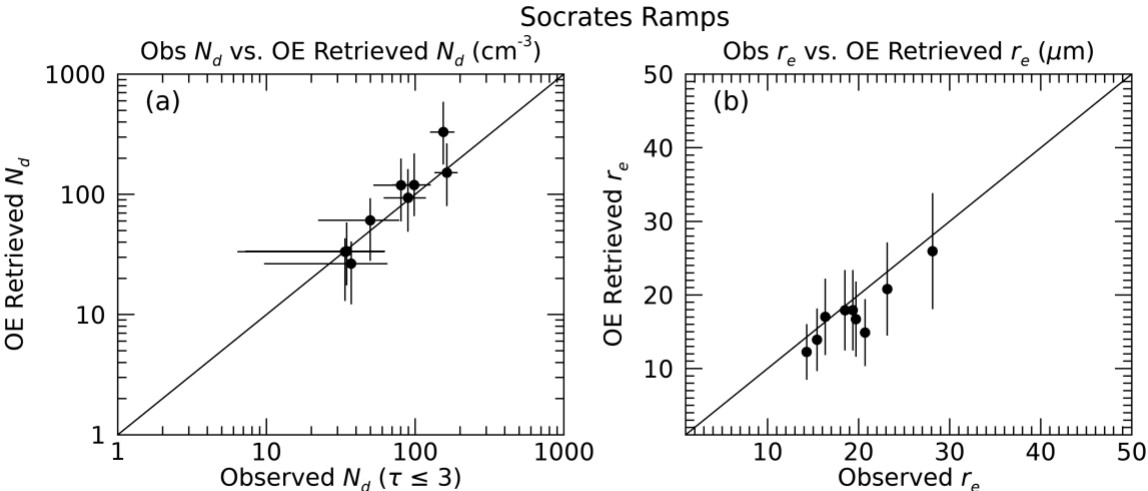

**Figure 5. Comparison of Observed and Derived $N_d$ (a) and $r_e$ (b) from Socrates ramps. The error bars on the retrieved quantities are as derived from the optimal estimation.**

### 3. Results and Discussion

In this section, we present independent comparisons of the results the Nd OE algorithm using a detailed case study collected when the Terra satellite passed over the RV Investigator approximately 1 hour prior to an in-situ sampling period conducted by

the NCAR GV during Socrates on February 18, 2018 (2/18). We then expand our view to examine comparisons of multiple overpasses of the ship by the NASA MODIS instrument on the Terra and Aqua satellites during the 2018 Summer campaigns.

## 3.1. A Case Study

The 2/18 case study provides a unique opportunity for independent comparisons of the algorithm with data collected while the GV aircraft operated in the vicinity of the RV Investigator and with an overpass of the Terra satellite that provided independent retrievals of $\tau$ and $r_e$ (Platnick et al., 2004) from which we can derive LWP and $N_d$ (G18) using the MODIS $\tau$ and $r_e$. During this case study period, the ship remained stationary at 56.6°S and 141.5°E to facilitate coordination with the GV. Figure 6 illustrates the data collected from the shipboard instruments. The lidar attenuated backscatter indicates a fully attenuating layer through the entire period. With a cloud base temperature near -5C, the lidar depolarization ratio data suggest that the cloud base phase and the sub cloud precipitation were liquid. The W-Band radar on the RV Investigator indicated episodic drizzle events of 10-20 minute duration roughly every hour, some of it rather heavy. Intervening periods without drizzle had radar reflectivity near the detection threshold of the radar (~-25 dBZe in the Capricorn 2 configuration). The radar and sounding data collected at the ship showed that the layer was topped by a strong marine inversion near 1.5 km in agreement with the GV ramp in Figure 4. The LWP was variable between 50-60 g m$^{-2}$ during periods without drizzle to values near 250 g m$^{-2}$ during periods of drizzle. The retrieved cloud properties varied depending on the proximity of a drizzle event. While the algorithm did not converge in regions of heavier drizzle, we find near the boundaries of several drizzle events that the $N_d$ decreased to 20-30 cm$^{-3}$ and $r_e$ increased to be more than 20 $\mu$m. Otherwise, the algorithm tended to produce $N_d$ in the range of 100 cm$^{-3}$ and $r_e$ in the 10 $\mu$m range.

A Terra MODIS overpass occurred at 0025 UTC. We collect the Level 2 retrieval of $\tau$ and $r_e$ in a region of 50 km diameter centered on the ship and the ship data are collected between 23 UTC on 17 February and 0130 UTC on 2/18. The comparison results are

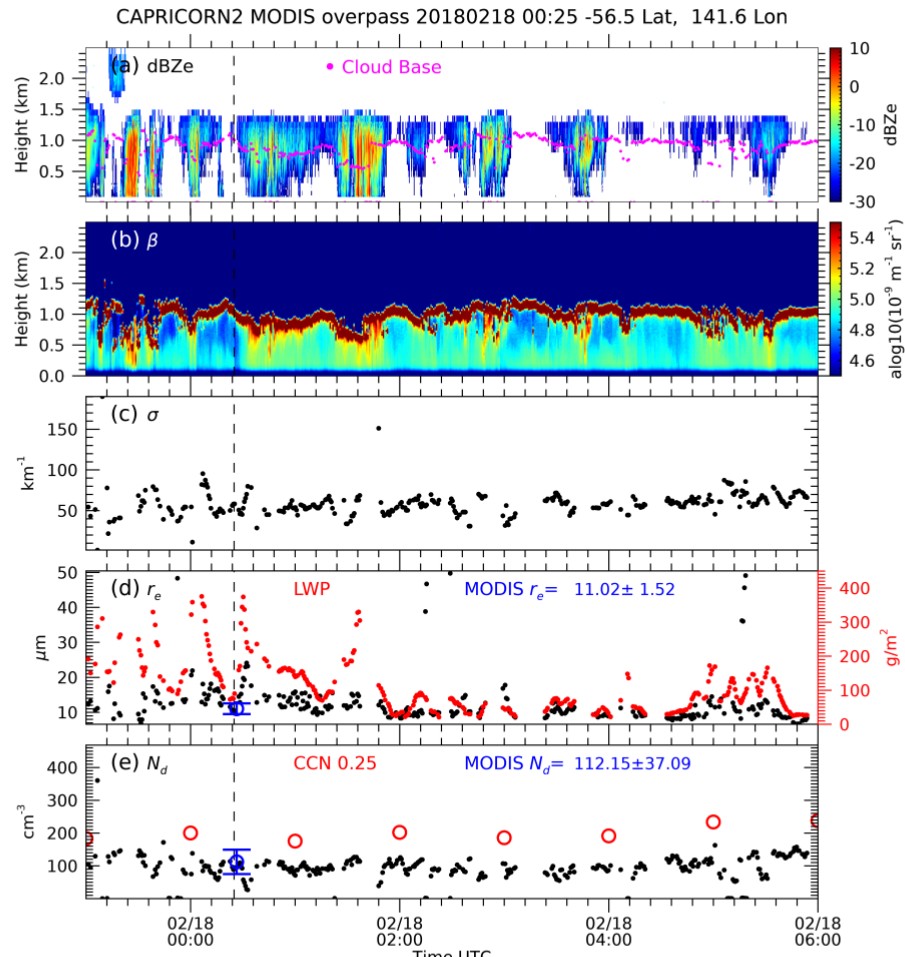

**Figure 6. Surface-based measurements and derived properties from data collected on February 18, 2018 on the RV Investigator near 55.6S and 141.5E. a) radar reflectivity ($Z$) with ceilometer cloud base indicated by purple dots, b) lidar attenuated backscatter, $\beta_{obs}$, c) extinction () derived from $\beta_{obs}$, d) $r_e$ and LWP, e) $N_d$. The blue circles in panels (d) and (e) and inset values are from an overpass at 0025 UTC (vertical dashed line) of MODIS on Terra. CCN at 0.25% supersaturation is shown on (e) using red circles.**

shown in Fig. 7 (see also Fig. 6). A broad distribution of LWP is demonstrated during this period that has a similar character in both data sets. The ship has an LWP mode near 160 g m$^{-2}$, that is due to the drizzle event that is evident near 00 UTC in Fig. 6. The mean LWP from the ship is slightly larger than MODIS but the two are in broad agreement. The distributions of $r_e$ in the two data sets overlap with the surface data skewed to larger values, likely because of the predominance of the drizzle event. The $N_d$ retrievals also demonstrate broad agreement with quite wide distributions even though the ship $N_d$ is skewed to smaller values. The ship $\tau$ distribution is skewed to smaller values than MODIS, consistent with larger $r_e$ and smaller $N_d$. We note that $\tau$ and $r_e$

are the quantities that are most directly retrieved from the MODIS algorithm, whereas the LWP and especially $N_d$ require additional assumptions.

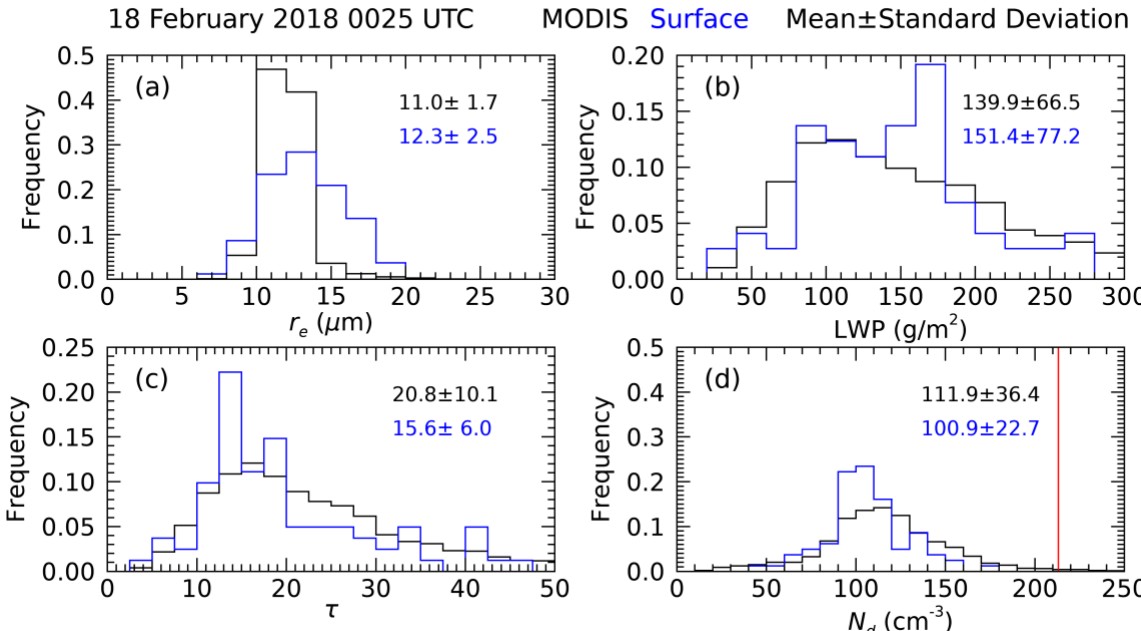

**Figure 7. Comparison of properties observed and derived from data collected on the RV Investigator (blue) with cloud properties derived from a Terra MODIS overpass at 00:25 UTC on February 18, 2018. a) $r_e$, b) LWP, c) Optical depth, d) $N_d$. The vertical red line on panel (d) shows the 0.25% supersaturation CCN measured on the RV Investigator at this time.**

On the other hand, the surface data LWP is independent of the radar, lidar, and other measurements and requires a minimum of assumptions to derive from the microwave radiometer brightness temperatures (Turner et al., 2016). $N_d$ and $r_e$ from the surface data require a complicated algorithm, and $\tau$ from the surface data is calculated using Eqn. 9. Thus, the surface-derived $\tau$ would include the errors in the surface retrieval of $r_e$. While there are biases in the comparison, given the substantial differences in the two independent data sets, we conclude that the comparisons demonstrate a reasonably consistent picture of the cloud field during the overpass.

The GV arrived at the ship at approximately 02 UTC on 2/18 and operated in the vicinity of the ship for roughly 2 hours. It conducted ramps, level legs within the cloud layer, and legs above and below the layer for aerosol and remote sensing applications.

We compare data collected during this time by gathering the aircraft data within 50 km of the ship. The effective radius is derived from the aircraft CDP data in the upper ½ of the layer (above 1.2 km) and the aircraft $N_d$ is collected from the CDP data in the lowest ½ of the layer. The comparison of $N_d$ and $r_e$ distributions are shown in Fig. 8. The aircraft $r_e$ data are bimodal while the ship retrieved $r_e$ are unimodal and centered on the lower mode of the aircraft $r_e$ distribution. We interpret the lack of bimodality in the ship-based $r_e$ data as being due to the algorithm not converging in regions of heavier drizzle as noted above. The aircraft penetrations of drizzle and non-precipitating clouds results in the bimodality in the $r_e$ distribution shown in Fig. 8. The $N_d$ distributions are broadly similar, but the ship results are biased to lower values. It is unclear the extent to which there is a bias toward the lower part of the cloud layer in the ship data. Regardless, both distributions are centered just in excess of 100 cm$^{-3}$. This comparison suggests that the surface-based OE algorithm reasonably replicates the cloud layer properties in this case.

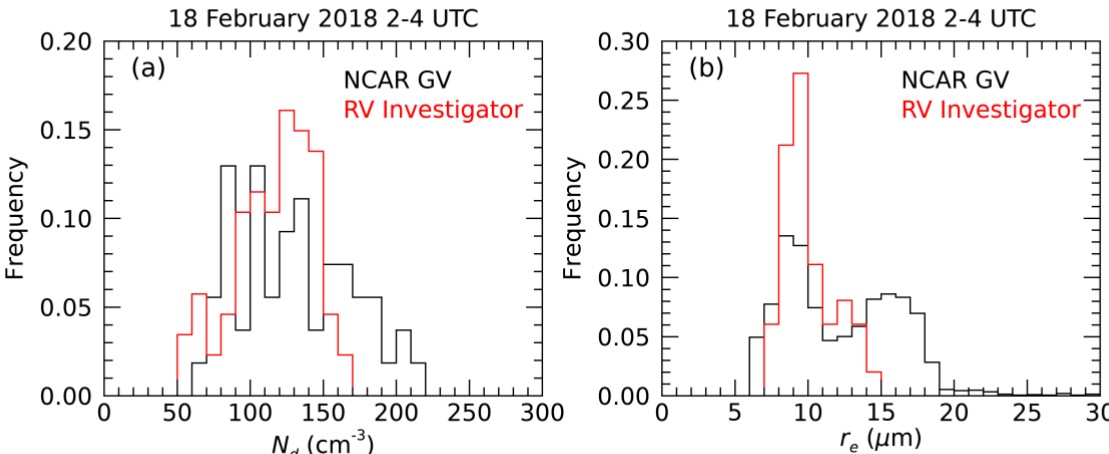

**Figure 8. Comparison of $N_d$ (a) and $r_e$ (b) derived from the surface-based data collected on the RV Investigator (red) with data collected from the NCAR GV on 18 February 2018. Cloud properties are compiled over the period from 2-4 UTC.**

Overall, we find that the aircraft, satellite, and surface-based data sources provide similar and very interesting characterizations of the cloud and CCN on 2/18. Twohy et al. (2021) in their supplemental information show that the airmass above the marine boundary layer on 2/18 had one of the highest sulfur-based concentrations of CCN recorded during Socrates at 224 cm$^{-3}$. The air mass observed on 2/18 followed a trajectory from the deep south over the Antarctic continent and the biologically productive waters of the Southern Ocean. The high concentrations of sulfate CCN in the free troposphere imply new particle formation along the trajectory was likely responsible for the high CCN (McCoy et al., 2021). The CCN at the surface measured on the RV Investigator was near 210 cm$^{-3}$ – slightly lower than that measured on the aircraft.

On the other hand, $N_d$ seems to be consistently in the 100 cm$^{-3}$ range from the surface, ship, and MODIS except for the near-cloud top maxima in $N_d$ observed by the GV in the ramp demonstrated in Fig. 4. The other ramps (not shown) also had values of $N_d$ near

the CCN values of 200-250 cm$^{-3}$. We speculate that the difference between CCN and $N_d$ is mostly likely due to precipitation droplet scavenging and coalescence process that is actively generating drizzle in this case. The high CCN from the free troposphere transported to this location from the south is likely mixing into the marine boundary layer through entrainment (the cloud top spike in $N_d$ in Fig. 4) and being processed through clouds explaining the lower surface CCN. The cloud properties ($N_d$ in the 100 cm$^{-3}$ range) are a drizzle and coalescence damped response to the higher free tropospheric CCN.

### 3.2. Expanded MODIS Comparison

Finally, we compare with the MODIS-derived cloud properties from overpasses of the ships during the Marcus and Capricorn campaigns. With MODIS instruments on the Terra and Aqua satellites and the ships being at sea over extended periods, we found

several events where suitable low-level clouds occurred over the ships during MODIS overpasses. Table 4 lists the information about the 14 overpasses of the ships that we use for the comparison in Fig. 9. Our approach was to examine a 50 km region of MODIS data centered on the ship, and we compiled surface data from 90-minute periods before and after an overpass. We find

reasonable agreement in the comparisons. The LWP is an interesting quantity since, as stated above, it is independent of the $N_d$ - $r_e$ retrieval. The LWP from the MODIS data, on the other hand, is derived from the $\tau$ and $r_e$ algorithm that uses a bi spectral method (Nakajima and King, 1990) so that the MODIS LWP would carry forward any uncertainties in $\tau$ and $r_e$. The agreement, however, is reasonable with little bias. Most of the cases have LWP<200 g m$^{-2}$ since we focus on non- to lightly precipitating cloud scenes. The $r_e$ of the cases range over values that are very small corresponding to cases near the Antarctic continent with high $N_d$ and no precipitation to $r_e$ that exceeds 15 $\mu$m. The comparison in $r_e$ is unbiased with a reasonable correlation. While $N_d$ also demonstrates reasonable correlation, there does appear to be a slight bias in the comparison, with the surface data being, on average, 20-30 cm$^{-3}$ higher than MODIS. The optical depth appears unbiased for values less than ~15 but then seem to show a bias for values of more than 15 with MODIS being larger than the surface-based results. More data is highly desirable to establish how well and under what circumstances these data sets agree or don't, but this comparison is encouraging.

**Table 4. List of the MODIS overpasses shown in Fig. 9.**

| Date/Time | Location | Satellite | Campaign |
|---|---|---|---|
| 2018/01/29, 0450 UTC | | Aqua | Capricorn 2 |
| 2018/02/04, 0415 UTC | 65.6°S, 150.0°E | Aqua | Capricorn 2 |
| 2018/02/05, 0455 UTC | 63.9°S, 150.0°E | Aqua | Capricorn 2 |
| 2018/02/07, 2350 UTC | 62.8°S, 143.6°E | Terra | Capricorn 2 |
| 2018/02/13, 0545 UTC | 63.9°S, 132.1°E | Aqua | Capricorn 2 |
| 2018/02/18, 0025 UTC | 56.5°S, 141.6°E | Terra | Capricorn 2 |
| 2018/02/20, 0010 UTC | 50.2°S, 143.7°E | Terra | Capricorn 2 |
| | | | |
| 2018/01/02, 0110 UTC | 66.3°S, 110.5°E | Terra | Marcus |
| 2018/01/05, 0140 UTC | 66.2°S, 110.2°E | Terra | Marcus |
| 2018/01/05, 0720 UTC | 66.1°S, 110.0°E | Aqua | Marcus |
| 2018/01/06, 0225 UTC | 64.0°S 111.3°E | Terra | Marcus |
| 2018/01/10, 0425 UTC | 47.0°S 142.6°E | Terra | Marcus |

| 2018/02/23, 0805 UTC | 59.3°S, 89.3°E | Aqua | Marcus |
|---|---|---|---|
| 2018/02/24, 0305 UTC | 56.9°S, 95.4°E | Terra | Marcus |

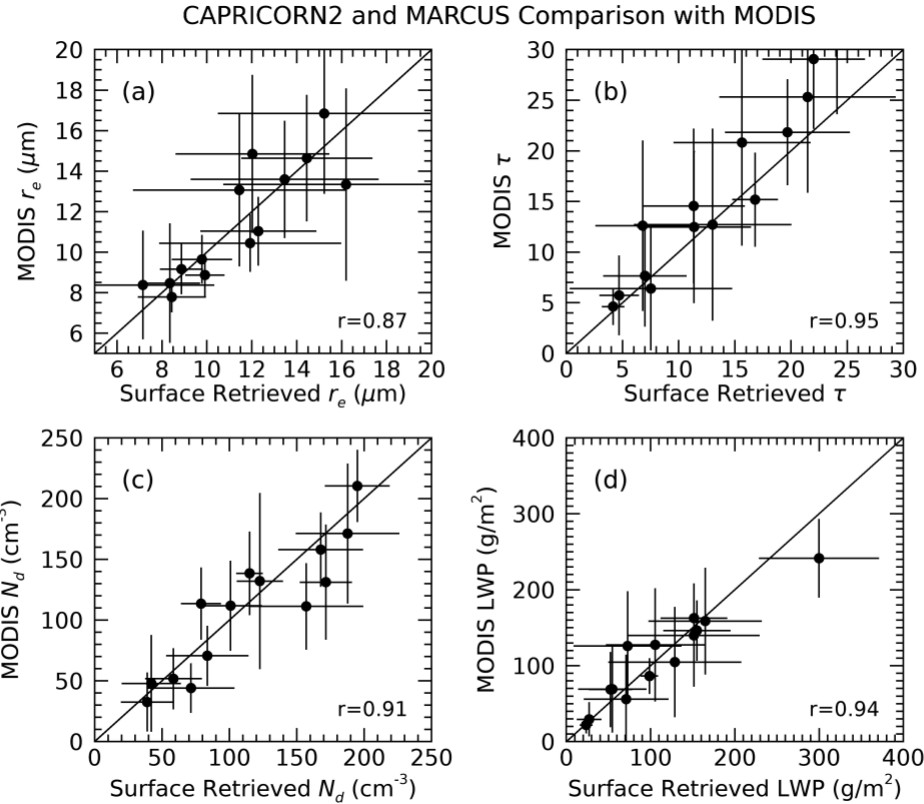

**Figure 9. Comparison of MODIS derived cloud properties with cloud properties derived from data collected during the Marcus and Capricorn 2 campaigns in the Southern Ocean during Austral Summer 2018. Error bars are 1 standard deviation of the retrieved cloud properties during the time and over the spatial extent of the two data sets. The Pearson correlation coefficient of the comparison is shown as an inset in each panel.**

### 5. Summary and Conclusions

Given the importance of knowing cloud droplet number concentrations ($N_d$) in low-level clouds for understanding how these clouds interact with aerosol and precipitation-producing processes to influence the earth's albedo, we have explored two techniques that allow us to derive $N_d$ and layer effective radius ($r_e$) using surface-based remote sensing techniques with an emphasis on the information brought to this problem by lidar data. The depth a laser pulse penetrates a cloud layer is a function of the amount of water droplet cross-sectional area presented to the laser pulse, and that cross-sectional area is dependent upon the $N_d$ and the liquid water content ($q$). This observable is quantified by the lidar attenuated backscatter, $\beta_{obs}$, (Eqn. 10) that is modulated by the directionality of the scattering as represented by the multiple scattering factor. As the lidar beam penetrates a cloud layer, the signal initially increases until two-way attenuation causes the signal to reach a maximum, after which it decays exponentially depending upon multiple scattering. The rate of increase in $\beta_{obs}$ is easily quantified if $N_d$ and $q$ are known, or turning the problem around,

480

one can calculate $N_d$ if $\beta_{obs}$ is observed, and $q$ is known. The math becomes more tractable where the lidar signal is at a maximum (a distance we term $R_{max}$) since the rate of change of $\beta_{obs}$ is zero there (Eqns. 16 and 17). The liquid water content, $q$, can be expressed in terms of the rate of increase of $q$ with height for an adiabatic cloud which can be made more realistic by scaling the $q$ profile by an adiabaticity factor that can be derived from LWP and cloud layer depth. This simple model (Eqn. 17) can be implemented with an estimated cloud depth, LWP, and a lidar. The effective radius near cloud top can then be derived (Eqn. 18).

The method, however, is very sensitive to uncertainty in $R_{max}$ which is, in turn, dependent on the vertical resolution of the lidar. Since $R_{max}$ typically ranges from a few 10's to maybe as much as 100 m, the uncertainty in derived $N_d$ becomes prohibitively large (> 100%) for range resolutions much above 15 m. $R_{max}$ also depends on an estimate of where in the vertical profile activation of cloud droplets begin. In non-precipitating clouds, this level is easily discerned to be where the signal first rises significantly above the aerosol and molecular background. In light precipitation, this level is less obvious, and we extrapolate the signal to a level of signal strength that was previously identified in non-precipitating conditions. We found empirically that the cloud base identified by most automated ceilometer or lidar algorithms typically identify a cloud base to be very near where the lidar attenuated backscatter reaches a maximum which is not useful in this context. Uncertainty in $R_{max}$ translates predictably into uncertainties in $r_e$. Another limitation of the method is the need to estimate the $q$ profile above cloud base. We take advantage of an assumed adiabatically shaped $q$ profile to estimate $q$ at the point where $\beta_{obs}$ reaches a maximum. This allows us to essentially have two pieces of information to solve Eqn. 2 with the third, $\alpha$, being assumed. A cloud that does not have this adiabatic shape in $q$ would, therefore, not provide an accurate estimate of $N_d$ and $r_e$. Additionally, the cloud must be fully attenuating to have an accurate value for $R_{max}$. We assume that most optically thick stratocumulus would satisfy these assumptions. Note that it would be difficult to adapt this method to down looking observing systems from aircraft or satellite because of the assumption of the adiabatic shape of the $q$ profile. The tops of many MBL clouds contain a region where $q$ is decreasing with height from a layer maximum $q$ due to interaction with dry air at the layer top. The depth of this region would depend on the strength of the marine inversion and the amount of mixing.

To lessen the effects of uncertainty in $R_{max}$, we bring more information to bear on the problem by quantifying the cloud layer extinction in terms of the rate of decay of the lidar signal beyond $R_{max}$ using a published methodology (Li et al., 2011). In addition, we use the radar reflectivity near cloud top as a constraint on the $q$ profile and $r_e$. This is cast in an optimal estimation (OE) algorithm that seeks to balance the uncertainty in the observations and uses prior information such as CCN concentrations that provide an upper limit on $N_d$. The OE algorithm is only marginally successful in reducing the uncertainty in $N_d$ and $r_e$. The uncertainties, especially on $N_d$, remain substantial since $R_{max}$ provides the most significant information on $N_d$ and the other measurements provide minimal constraint on $N_d$ as quantified in the Jacobian ($K_x$) matrix. What we find interesting but not surprising is that the use of CCN as a prior constraint allows us to balance the information content in $R_{max}$ and the other observations with what we know as a significant constraint on $N_d$ and, to a lesser extent, $r_e$. Overall, the OE uncertainties that are shown to be reasonable through a bootstrapping experiment and through comparison to aircraft data, are in the range of just under a factor of 2 for $N_d$ and 30% for $r_e$ for lidar range bins of 10-30 m. The only way to reduce this uncertainty is to have dedicated lidar measurements that have vertical resolution less than 10 m. Using comparisons with in-situ aircraft data and with cloud properties derived from MODIS, we show that the OE algorithm provides results consistent with the uncertainty in the data and retrievals.

Finally, a case study is explored that shows how synergistic remote sensing data from the surface, especially when combined with aircraft and satellite data, can be exploited. The February 18, 2018 case study that took place in the Southern Ocean near 56°S and

141°E suggests how long-range aerosol transport of an air mass from the biologically productive waters of the deep southern latitudes modulated the cloud properties that existed on this day. The CCN measured at the surface and from the GV aircraft was about a factor of two larger than the ~100 cm$^{-3}$ $N_d$ inferred from the ship-based remote sensing and MODIS data and observed by the GV. This difference between $N_d$ and CCN was likely a response to the widespread precipitation processes that were occurring on this day.

## 6. Competing Interests

The author declares no conflict of interest.

## 7. Code and Data Availability

All data used in this study are available in public archives. MODIS cloud products can be found for Terra and Aqua at https:/doi.org/10.5067/TERRA/MODIS/L3M/CHL/2018 and http://dx.doi.org/10.5067/MODIS/MYD06_L2.006. ARM data can be obtained at https://www.arm.gov/data/. SOCRATES data are available at https://data.eol.ucar.edu/project/SOCRATES, CAPRICORN 2 data are available at https://doi.org/10.25919/5f688fcc97166. Computer code for this study including all analysis code and graphic generation code is written in the IDL language. Code is available upon request to the corresponding author.

## 8. Acknowledgments

This work was supported by NASA Grant 80NSSC21k1969, DOE ASR Grants DE-SC00222001 and NSF Grant 2246488. The author would like to thank Sally Benson for her expertise in adapting code and generating figures. Alain Protat enabled the author's participation in the CAPRICORN 2 campaign and provided the cloud radar and lidar data from the RV Investigator. Ruhi Humphries provided the CCN data filtered for ship exhaust. We thank Dr. Connor Flynn for helpful discussions regarding interpretation of the Micropulse lidar data from Marcus. This Research was supported, and data were obtained from the Atmospheric Radiation Measurement (ARM) User Facility, a U.S. Department of Energy (DOE) Office of Science user facility managed by the Biological and Environmental Research Program. Technical, logistical, and ship support for MARCUS were provided by the Australian Antarctic Division through Australia Antarctic Science projects 4292 and 4387 and we thank Steven Whiteside, Lloyd Symonds, Rick van den Enden, Peter de Vries, Chris Young and Chris Richards for assistance. The author would like to thank the staff of the Australian Marine National Facility for providing the infrastructure and logistical and financial support for the voyages of the RV Investigator. Funding for these voyages was provided by the Australian Government. All data used in this study are available in public archives.

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
