# Peer review of "Deriving cloud droplet number concentration from surface based remote sensors with an emphasis on lidar measurements."

_EGUsphere, 2023_

## Referee Comment (RC2)

**Review of "Deriving cloud droplet number concentration from surface based remote sensors with an emphasis on lidar measurements" by Gerald Mace**

The manuscript presents an original and ingenious approach that uses a combination of common ground-based remote-sensing instruments (ceilometer, cloud radar, and microwave radiometer) for inferring the effective radius ($r_e$) and number concentration ($N_d$) of cloud droplets in geometrically thin low-level liquid water clouds over the Southern Ocean. The new method is centred around what's usually a nuisance to lidar operators: the full attenuation of the laser beam in optically thick clouds. The author expands on earlier work to establish a straightforward physical connection between the distance between cloud base to full attenuation of the lidar signal and $N_d$ (and thus $r_e$). The new framework requires a few assumptions as well as auxiliary cloud information from measurements with cloud radar and microwave radiometer.

The author finds that the application of the new method is strongly constrained by the range-resolution of common ceilometers and lidars, as this determines the accuracy to which the distance between cloud base to full signal attenuation can be determined. The authors hence developed an optimal estimation retrieval based on the novel framework to incorporate more physically meaningful information from the other remote-sensing instruments into the analysis.

The results are then evaluated through a detailed case study that employs independent airborne in-situ measurements and satellite observations as well as through comparison to $N_d$ and $r_e$ from MODIS observations.

The work is an excellent fit for AMT. However, I believe that major revisions are needed before the work can be published as outlined in my comments below.

I want to disclose that I had some editor-mediated communication with the author to clarify the derivation of Eq. (9). While some of the equations in the manuscript are erroneous, the author has confirmed that their implementation in the analysis routine is correct. I could now follow the presented theory. However, there still seem to be some typos that I will point out below as well.

**Major comments:**
- **Structure:** I suggest to restructure the manuscript to make it easier for readers to follow the process of development, application, and evaluation. For instance, I suggest to introduce all considered measurements as well as the data comparison strategy before the new method is described. At least for me, knowing what data will or might be used really supports the thought process. It also enables a much more straightforward presentation of the findings later in the manuscript. Here is a potential structure:
  1 Introduction
  2 Data and methods
  2.1 Instruments and comparison approach
  2.2 Theory
  2.3 An optical estimation algorithm
  2.4 OE evaluation
  3. Results and discussion
  3.1 Detailed case study
  3.2 Long-term comparison
  4. Summary and conclusion.

- **Variables:** I advise the re-evaluate the choice of variable names and to check for their consistency throughout the manuscript.

  - The lidar equation uses both $z$ and $r$ for distance. $z$ might be confused with $Z$, $r$ might be confused with droplet radius. I find the potential for a mix-up between $r_e$ and $r_{max}$ (zmax is also used) particularly problematic. Possible solutions include a list of symbols (which might be too extensive for this work) or the use of different signs for distance or $r_{max}$.

  - The author switches between different names and signs, particularly between the text and the figures. For instance, liquid water content is both $LWC$ and $q$. The cloud droplet number concentration is $N_d$, Nd, or cloud droplet number. The cloud droplet effective radius is $r_e$, re, Re, or effective radius. I suggest to introduce the formula signs and stick to them throughout the manuscript and the figures. There is no need to re-introduce them in the summary.

- **Figures:** Rather than using figure and panel titles, I suggest to provide a full description of a figure in the figure caption. This includes what's shown in the different panels (and in which colour), the time and location of measurements, and the source if measurements are shown.

- **Derivation of Eq. (9):** It would be easier to outline the derivation of Eq. (9) if each of the equations in Eq. (2) and (3) had their own number. I can only arrive at the authors derivation if the following equations are corrected.
  Eq. (2) should have a denominator of 2:

$$\sigma = \frac{\pi}{2} N_0 D_0^3 \Gamma(\alpha + 3) \,.$$

Eq. (A1), which is a re-arranged version of Eq. (2.4) with Eq. (2.1) substituted for $N_0$, should have only the second power of $D_0$:

$$\sigma = \frac{\pi}{2} \frac{N_d}{\Gamma(\alpha + 1)} D_0^2 \Gamma(\alpha + 3) \,.$$

The combination of the equation for $D_0$ and $q$ (Eq. (2.2)), which has no number in the Appendix should feature the term $\Gamma(\alpha + 4)$ from the expression for $q$:

$$D_0 = \frac{3}{\rho} \frac{q}{\sigma} \frac{\Gamma(\alpha + 3)}{\Gamma(\alpha + 4)} \,.$$

This leads to a corrected equation for $B$ in Eq. (A2):

$$B = \left( \frac{9\pi}{2\rho^2} \frac{\Gamma(\alpha + 3)^3}{\Gamma(\alpha + 4)^2 \Gamma(\alpha + 1)} \right)^{\frac{1}{3}} \,.$$

Eq. (6) should feature $\sigma$ rather than $r$ in the second term on the right side:

$$\frac{\partial \ln \beta_{obs}}{\partial r} = \frac{\partial \ln \sigma}{\partial r} - 2\eta\sigma \,.$$

The $B$ in Eq. (A4) should not be to the power of 3, while the entire Eqs. (7,8) should be as already stated in the clarification. I might also be wrong so please double-check.

- **Measurement uncertainty:** I am trying to wrap my head around the reasoning in the lidar range-gate spacing and the subsequent discussion of Figure 3. I understand that finer range resolution allows for a better quantification of $r_{\max}$ as it provides to better resolve where exactly the lidar signal becomes saturated. If I have a common range bin of 15 m and my nominal height is at the bin centre, wouldn't it mean that my range uncertainty is 7.5 m rather than the full 15 m? I wonder if measurements capabilities are actually a factor 2 better than currently accounted for?
  After the discussion of Figure 3, it is concluded that the $r_{\max}$ as inferable from the lidar measurements is insufficient to retrieve $N_{\mathrm{d}}$ and $r_{\mathrm{e}}$ and that the use of additional information as in the optimal inversion algorithm could compensate for this lack. Later, all presented results are inferred by the OE algorithm. I wonder what the results would look like if the author had used the lidar-derived $r_{\max}$. Could you please comment or provide an example?

- **Verification:** I suggest to restructure the verification section of the paper. The airborne in-situ measurements allow for a more direct comparison then the MODIS observations. I would therefore discuss those (they would already be introduced in Section 2) first in the detailed case study (currently lines 374 - 386), continue with the addition of MODIS observations during the case study period (currently lines 358 - 373), and conclude with the comparison to the other coinciding MODIS overpasses (currently lines 387 onward).

- **Bigger picture:** I find this to be an excellent paper that could be even better if the author could sort the new method into the bigger picture of cloud remote sensing. You outline what kind of measurements are needed. Maybe you could also provide a check list on ranges of cloud properties for which you would assume the method to be valid? Is it just shallow liquid water clouds that don't precipitate too heavily? Are there other regions or established measurement sites for which the method could be applied? Would such an application require a revision of the OE algorithm? Which knowledge gaps could be closed by a widespread application of the new method?

**Specific comments:**
- line 174: please give all normalised standard deviations in either percent or without units

- line 252: to wit?

- equation for $K_{\mathrm{b}}$: second row, third column should be $\partial\sigma/\partial\eta$

- line 275: I suggest to start the section on the evaluation of the OE algorithm here as the text before is still on the functioning of the algorithm

- Tables 2 and 3: Is it possible to combine those table, maybe in a transposed form? I also suggest to include cases 2, 4, and 6 in Table 2 as this is what's stated in the text. Please also check the referencing between the tables if kept separated.

- line 194: not clear how this is shown in Table 2

- Figure 6: Maybe show plots for radar and lidar to 2 km height only?

- Figure 7 and in the text: integers should suffice for $N_{\mathrm{d}}$

- line 397: please quantify what is meant with good correlation

- Table 4 could be moved to the appendix

- Figure 9: It's hard to be certain but I have the impression that there are more points in the plots than overpasses listed in Table 4...?

---

## Author Comment (AC1)

I appreciate the review of this paper provided by Dr. Baumgardner. He raises several important points and I have tried to address them in my reply and with additional or clarifying language in the manuscript. In the following, I copy each of his comments and follow that with a reply.

- Line 76: "Nd (re) has uncertainties of 0.16(0.16) and 0.55 (0.18), respectively." What are the units/dimensions of these uncertainties? They can't be in units of cm-4 and they are too small to be percentages. Can you explain?

Reply: These are fractional uncertainties that would be realized in the idealized situation where we have 1 meter and 5 meter lidar vertical resolution. I agree that the values for 1 m are optimistic since other uncertainties would become dominant. But a 55% uncertainty at 5 meters for Nd seems reasonable in my mind and somewhat startling since most lidars don't collect data at this resolution. This would mean that for a 100 cm$^{-3}$ value for Nd the error bar would range between 55 and 155. The cloud top effective radius uncertainty on the order of 15-20% is quite substantial given the typical dynamic range of this quantity (8-15 microns for most non precipitating liquid clouds). I've attempted to add a bit more nuance to the text to explain this.

- Line 211: "Observed thermodynamics". I am assuming that this refers to soundings that document the T/RH vertical profiles? I return to this question further down when I ask for more information on how the uncertainty in this profile impacts the derived Nd.

Reply: Correct. I have added a bit of clarifying text to this statement in the revision.

- Line 229: "For the prior estimate of Nd, we reason that coincident cloud condensation nuclei (CCN) measurements provide an upper limit on the droplet number in each situation.". This requires additional discussion because the CCN measurements are only relevant as an upper limit to Nd if the cloud base temperature and maximum updraft velocity is known, since the number of droplets activated depend on the CCN spectrum of concentration versus supersaturation (SS). Two sentence further a value of 0.2% was mentioned, but where did this number come from? Unless I overlooked it, no where in the article are vertical motions discussed. Yes, given the cloud base T/P you can estimate maximum LWC but not maximum % SS. Given the very nice correlation between the in situ measured Nds and those extracted from the remote sensors, maybe this is a moot point. Perhaps 0.2% is a good guess for the clouds studied on the Southern Ocean; regardless, a bit more discussion about the properties of the CCN in this region would be useful with regard to the conditions that activate them.

Reply: We found that the use of CCN as a broad constraint on the iterative algorithm was necessary to achieve convergence and agreement with independent data. As we point out in the discussion of the Kx matrix (Jacobian) and elsewhere in the paper, the amount of information from the lidar regarding Nd is just barely sufficient for our purpose and CCN provides a useful and necessary starting point. I chose to use CCN at 0.2% based on a figure in McFarquhuar et al., 2021(Figure 5) and from discussions with colleagues who have worked with the Socrates data. We also performed our own analysis of the cloud probe and CCN data and found that 0.2% had slightly better correlation than 0.3. Of course, at a microphysical level the magnitude of the updraft also matters as you point out. However, the updraft velocity is a quantity that we cannot

know form surface-based remote sensing at sea.  However, I don't think this is a critical point.  We simply require a consistent starting point for the iterative algorithm.  The CCN are always an upper limit on Nd and most of the time the algorithm converges on a value 2/3 or ½ of the CCN especially in drizzle.  I have added clarifying text to the revision on this point.

- Figure 4 and line 316. "Ramp" is a term that I rarely see when aircraft measurements are conducted. I understand the intent but thin a single sentence that explain that a "ramp" is when the aircraft does a vertical profile through a cloud. What isn't clear is if these are multiple passes through a cloud at different altitudes or a constant climb or descent?

Reply:  Clarified in the revision.  These are approximately constant rate climbs or descents from base to top.

- When looking at the uncertainty analysis, I was unable to tell if uncertainty in fad and the associated uncertainty in derived adiabatic liquid water is included. The adiabatic LWC is sensitive to the LCL, i.e. cloud base temperature and pressure. These can vary throughout the day and even from initial values derived from radiosondes. How does this uncertainty impact the subsequent uncertainty in Nd and re?

Reply:  Yes, these are factored in via the adiabatic liquid water lapse rate.  See this in the expanded derivation that I included in the revision just below what is now equation 15. The adiabaticity of the layer is a critical parameter since it essentially tells the algorithm what the liquid water content (LWC or q) is at the point where the lidar attenuated backscatter reaches a maximum.  This $2^{nd}$ piece of information, the LWC, allows us to solve the equation for Nd.

- My last point is trivial but from my perspective as one who provides the community with instrumentation I would like the model and manufacturer listed along with the instrument discussed. The author explicitly mentions the Vaisala ceilometer but not the manufacturer or models of the micropulse lidar (MPL) and cloud droplet probe (CDP). I ask that these are added.

Reply:  I appreciate the need for this.  I have added this information to the extent I could find it in publications and reports.  Where I wasn't sure, I added references.  We could not do the science without these advanced instruments even though sometimes we errantly take the effort of the developers of these instruments for granted.

---

## Author Comment (AC3)

I would like to thank Dr. Tesche for his thorough review of my initially submitted manuscript. The paper is immeasurably better because of his effort. I have addressed each of the suggestions in the manuscript and I provided written responses below.

Major comments:

- **Structure**: I suggest to restructure the manuscript to make it easier for readers to follow the process of development, application, and evaluation. For instance, I suggest to introduce all considered measurements as well as the data comparison strategy before the new method is described. At least for me, knowing what data will or might be used really supports the thought process. It also enables a much more straightforward presentation of the findings later in the manuscript. Here is a potential structure:

Reply: I have taken the reviewers comments seriously and restructured the manuscript largely in accordance with this Reviewers suggestion although I did not follow it verbatim. I did add an introductory paragraph as suggested that will hopefully allow the readers to better know what instruments and data are being used and how we will combine them and why. I have also restructured the results section as suggested. I think that this new organization makes good sense and I appreciate this Reviewer's suggestion.

- **Figures:** Rather than using figure and panel titles, I suggest to provide a full description of a figure in the figure caption. This includes what's shown in the different panels (and in which colour), the time and location of measurements, and the source if measurements are shown.

Reply: We have reworked the figure captions as suggested.

- **Derivation of Eq. (9):**

Reply: I am very appreciative of the time this Reviewer took in going through the math. I've made all the changes suggested after finding my transposition errors from my handwritten notes. I have numbered all relevant equations. Instead of including an appendix, I decided to just include the detailed derivation of what was equation 9 in the body of the text. I think this makes more sense and allows a reader to follow the derivation easily.

- **Measurement uncertainty:** I am trying to wrap my head around the reasoning in the lidar range-gate spacing and the subsequent discussion of Figure 3. I understand that finer range resolution allows for a better quantification of rmax as it provides to better resolve where exactly the lidar signal becomes saturated. If I have a common range bin of 15m and my nominal height is at the bin centre, wouldn't it mean that my range uncertainty is 7.5m rather than the full 15m? I wonder if measurements capabilities are actually a factor 2 better than currently accounted for? After the discussion of Figure 3, it is concluded that the rmax as inferable from the lidar measurements is insufficient to retrieve Nd and re and that the use of additional information as in the optimal inversion algorithm could compensate for this lack. Later, all presented results are inferred by the OE algorithm. I

wonder what the results would look like if the author had used the lidar-derived rmax. Could you please comment or provide an example?

Reply: The discussion presented by the reviewer is correct. In calculating the uncertainty that led to equation 3 I used a Gaussian PDF that had 1 sigma values of 1, 5, 15 and 30. That would correspond to range bin spacing of twicer the 1 sigma values. I have corrected the notation and the discussion throughout the text. While the results are more optimisitc, the magnitude of the uncertainty for range bins spacing of 10-20 m increases from 55% to 130%. Most operational lidars would have factor of two uncertainties in Nd which is marginally informative. Regarding the second half of this comment, I did run the direct calculation of Nd when I ran the OE. The results were noiser but seemed statistically similar. This is expected since Rmax is the primary source of information. Note also that I've changed the notation for range in the revision. I now use a capital R to differentiate it from effective radius.

- **Verification:** I suggest to restructure the verification section of the paper. The airborne in-situ measurements allow for a more direct comparison then the MODIS observations. I would therefore discuss those (they would already be introduced in Section 2) first in the detailed case study (currently lines 374 - 386), continue with the addition of MODIS observations during the case study period (currently lines 358 - 373), and conclude with the comparison to the other coinciding MODIS overpasses (currently lines 387 onward).

Reply: I did restructure the Results section of the paper – perhaps not precisely as suggested here but I took the Reviewer's comment seriously and moved the case study discussion from its own section into the section on the case study comparison/validation.

- **Bigger picture**: I find this to be an excellent paper that could be even better if the author could sort the new method into the bigger picture of cloud remote sensing. You outline what kind of measurements are needed. Maybe you could also provide a check list on ranges of cloud properties for which you would assume the method to be valid? Is it just shallow liquid water clouds that don't precipitate too heavily? Are there other regions or established measurement sites for which the method could be applied? Would such an application require a revision of the OE algorithm? Which knowledge gaps could be closed by a widespread application of the new method?

Reply: I added a paragraph to the conclusions section addressing many of these questions. I avoided a few of them such as the last one about wide-spread application. Obviously as the developer, I'd like to see the method applied widely since I developed it to be easily used. However, before I start pushing this idea, I'd like to mature the algorithm more and see if it holds up against wider tests. It is certainly not a panacea given the limited information we have in the measurements to retrieve Nd. I honestly think this type of method is about the best we can do but only more critical testing will tell if it is sufficient.

Specific Comments:

- line 174: please give all normalised standard deviations in either percent or without units

Reply:  Addressed

- line 252: to wit?

Reply:  I think my usage is correct.  The phrase to wit is followed by an example of something that has already been said – according to the online sources I looked at.  Here I'm providing the regression that I described in the previous part of the sentence.  Admittedly, I'm not an expert writer but I think this is ok.

- equation for Kb: second row, third column should be $\partial\sigma/\partial\eta$

Reply: Fixed

- line 275: I suggest to start the section on the evaluation of the OE algorithm here as the text before is still on the functioning of the algorithm

Reply: Agree.  Done.

- Tables 2 and 3: Is it possible to combine those table, maybe in a transposed form? I also suggest to include cases 2, 4, and 6 in Table 2 as this is what's stated in the text. Please also check the referencing between the tables if kept separated.

Reply:  I combined Tables 2 and 3 into a single table as suggested.

- line 194: not clear how this is shown in Table 2

Reply:  Table 1 shows the correlation coefficients among the off diagonal elements.  We use that correlation coefficient and the variance of the quantities to calculate the covariance.  Text is clarified.

- Figure 6: Maybe show plots for radar and lidar to 2 km height only?

Reply:  Done

- Figure 7 and in the text: integers should suffice for Nd

Reply:  Noted.

- line 397: please quantify what is meant with good correlation

Reply:  correlation coefficients are now included as an inset in Figure 7.

- Table 4 could be moved to the appendix

Reply:  I suppose so, but I think I'll choose to just keep it in the text.

- Figure 9: It's hard to be certain but I have the impression that there are more points in the plots than overpasses listed in Table 4...?

Reply:  Thanks for catching.  I did omit 2 overpasses from the table.  They are now included.  14 points in the table and figures.